# Digital multiplex ligation assay for highly multiplexed screening of β-lactamase-encoding genes in bacterial isolates

Manu Tamminen[1✉], Jenny Spaak[2], Lea Caduff[2], Hanna Schiff[3], Ramon Lang[3], Steven Schmid[3], Maria Camila Montealegre[2] & Timothy R. Julian [2,4,5✉]

Increasing incidence of antibiotic resistance in clinical and environmental settings calls for increased scalability in their surveillance. Current screening technologies are limited by the number of samples and genes that can easily be screened. We demonstrate here digital multiplex ligation assay (dMLA) as a low-cost targeted genomic detection workflow capable of highly-parallel screening of bacterial isolates for multiple target gene regions simultaneously. Here, dMLA is used for simultaneous detection of 1187 β-lactamase-encoding genes, including extended spectrum β-lactamase (ESBL) genes, in 74 bacterial isolates. We demonstrate dMLA as a light-weight and cost-efficient workflow which provides a highly scalable tool for antimicrobial resistance surveillance and is also adaptable to genetic screening applications beyond antibiotic resistance.

[1] University of Turku, 20014 Turku, Finland. [2] Eawag, Swiss Federal Institute of Aquatic Science and Technology, CH-8600 Dübendorf, Switzerland. [3] ETH Zürich, CH-8092 Zürich, Switzerland. [4] Swiss Tropical and Public Health Institute, CH-4002 Basel, Switzerland. [5] University of Basel, CH-4003 Basel, Switzerland. ✉email: manu.tamminen@utu.fi; tim.julian@eawag.ch

Antibiotic resistant bacteria are a global threat to human health, animal health and the environment[1]. β-lactamase producing antibiotic resistant bacteria are amongst the most concerning since they confer resistance to essential medicines[2,3]; and increase the likelihood of treatment failure and mortality relative to susceptible infections[4,5]. In community settings, carriage rates of extended-spectrum β-lactamase producing Enterobacteriaceae (ESBL-E), an important subset of β-lactamase producing antibiotic resistant bacteria, is increasing at particularly alarming rates in low and middle income countries[6,7].

Detailed antibiotic resistance gene (ARG) surveillance at regional, national, and global scale is required for the development of strategies aiming to mitigate this health threat. Current methods include PCR, multiplexed PCR[8], integrated PCR and sequencing[9], whole-genome sequencing[9,10], SmartChip™ qPCR[11] and targeted metagenomics[12]. High diversity of β-lactamase ARGs[13] impacts the applicability of the current methods for their detection and classification, resulting in limited scalability and relatively high per-sample costs. A scalable, low cost method is therefore needed for specific detection of ARGs to aid surveillance.

Here, we present a digital multiplex ligation assay (dMLA) for simultaneous detection of 1187 β-lactamase-encoding genes within 74 bacterial isolates. The assay integrates in silico probe design with ligation-based genomic target detection and next-generation DNA sequencing (NGS). Recently, a closely related approach, digitalMLPA, was applied for the detection of copy number alterations in T- and B-cell lymphoblastic leukaemia[14] and multiple myeloma[15]. A closely related approach, TAC-seq, was also recently introduced to improve the sensitivity of non-invasive prenatal trisomy testing[16]. Our dMLA assay is readily adaptable for detection of any number of ARGs and is equally applicable for any other genomic targets beyond ARGs. High scalability of dMLA in terms of sample throughput as well as ARG targets allows unprecedented insights into ARG epidemiology.

## Results

**dMLA combines ligation assays with NGS for high scalability.** In dMLA, two target specific half-probes (20 bp) are designed to anneal to adjacent regions on a target gene sequence within the extracted DNA of a bacterial isolate (Fig. 1a). Each of the two half-probes (right and left) include an overhang with a PCR binding sequence. The right overhang also includes a molecular barcode between the PCR binding site and the target binding site, which is used to differentiate between ligation events and act as a proxy for individual DNA fragments in the original sample. When left and right probes anneal, ligase is used to form a phosphodiester bond in between, allowing subsequent PCR amplification of the product. The PCR step simultaneously attaches a sample barcode as an overhang on the left PCR primer (Fig. 1b, c). Sample barcodes allow multiple samples to be pooled for NGS, with differentiation after sequencing the pooled amplicons (Fig. 1d). Probe sequencing data is transformed into molecular target counts by enumerating the number of individual molecular barcodes for each target using an in silico workflow (Fig. 1e).

**dMLA probe design utilizes in silico analysis for maximal coverage.** We prepared dMLA targets for 1187 of the 1557 β-lactamase genes in the Lahey dataset[3]. This subset was selected by detecting the universally conserved sequence motifs within the dataset using a BLAST-like algorithm[17], and subsequently using a set coverage algorithm to identify those motifs which cover 76 % of the set with 36 probes (Fig. 2, Table S1).

**dMLA permits multiplexed detection of ARGs in a large number of samples.** We demonstrated the utility of dMLA using genomic DNA isolated from representatives of common pathogens belonging to families *Alcaligenaceae*, *Enterobacteriaceae*, *Flavobacteriaceae*, *Moraxellaceae* and *Pseudomonadaceae*. Within 74 bacterial isolates screened, dMLA detected 194 β-lactamase-encoding genes with no false positives or negatives (Fig. 3, Supplementary Fig. 1). In addition to the ESBL-encoding genes detected and confirmed by PCR, intrinsic resistance genes that were universally detected were the chromosomally encoded AmpC-type β-lactamases $bla_{ADC}$, $bla_{PDC}$, and $bla_{MIR}$, detected in all (and only) the 17 *Acinetobacter baumannii*, 12 *Pseudomonas aerugonisa*, and one *Enterobacter cloacae* isolates, respectively, as well as the chromosomally-encoded oxacillinase $bla_{OXA-51-family}$ in all *A. baumannii* group isolates[18–22]. dMLA also detected the

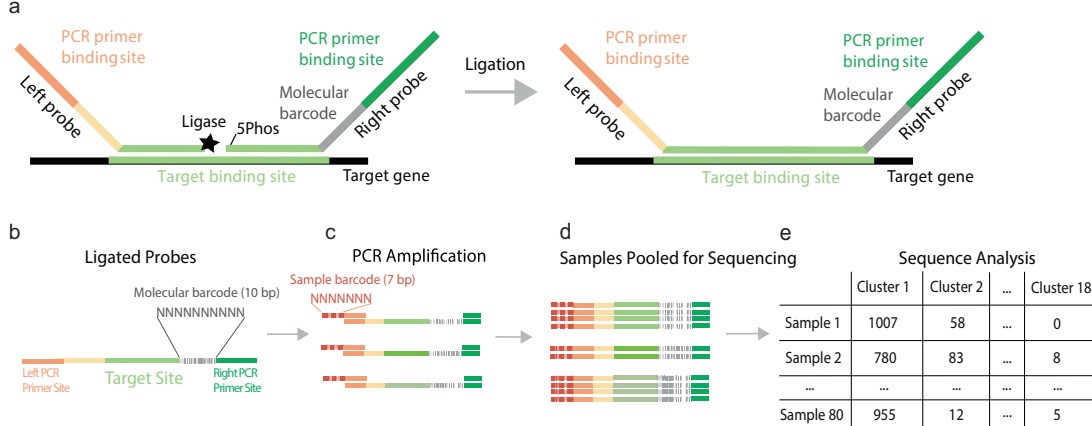

**Fig. 1 dMLA probe setup. a** Two adjacent half probes hybridized to target gene. Ligase connects the 3′ end of the left probe with the 5′ phosphorylated end of the right probe and the resulting fragment is amplified by conventional PCR. Molecular barcodes permit quantifying the number of target molecules while sample barcodes added during PCR permit pooling multiple samples on a single sequencing run. **b** Location of the molecular barcode sequences, target binding site, and PCR primer sites on post-ligation probes. **c** Multiple ligated probes within a sample with unique molecular barcodes are then amplified with PCR, which also attaches a unique sample barcode identifier. The primer binding sites are the same for all probes, so PCR amplifies any probes that ligate in the presence of a target gene region. **d** PCR products of amplified, ligated probes are then pooled for sequencing. **e** Sequencing reads are analyzed by the bioinformatics pipeline through sample and molecular barcodes.

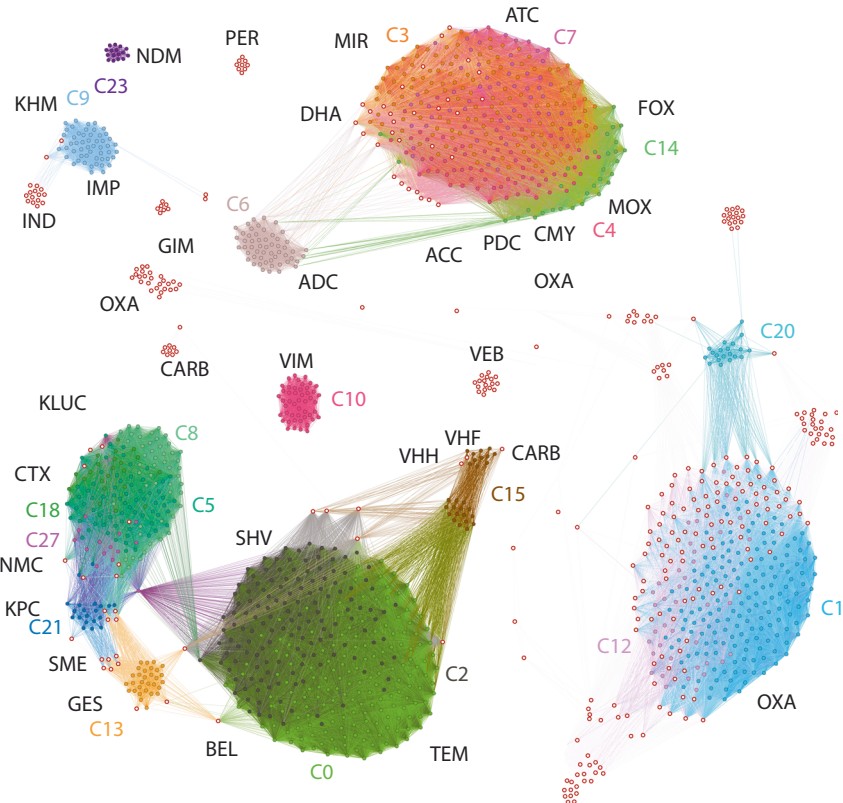

**Fig. 2 Network visualization of the dMLA probe coverage of the Lahey ESBL resistance gene dataset.** The nodes indicate individual resistance gene sequences and the edges at least one shared 40 bp sequence motif between them. The colours and coloured text indicate our probe designs, which cover particular clusters within the β-lactamase sequence pool, while the open circles with red outline indicate sequences for which probes have not been designed. The black texts indicate different β-lactamase clusters within the Lahey resistance gene dataset.

chromosomally-encoded *bla*_SHV_ in all five *Klebsiella pneumoniae* isolates[23].

## Discussion

Here, dMLA was specifically designed for highly parallel screening of bacterial isolates for β-lactamase-encoding genes within the context of national, regional, or global surveillance programs. Detection of intrinsic resistance genes highlights the specificity achievable with the dMLA method. Notably, the combination of the number of probe pairs and bacterial isolates demonstrated here is not reflective of the upper limits of the assay. dMLA could support substantially higher numbers of both probe pairs and/or bacterial isolates, with limitations theoretically driven only by sequencing depth. Additional probe pairs may be beneficial, particularly to highlight clinical relevance. As an example, many of the *Acinetobacter baumannii-calcoaceticus*-complex isolates contained oxacillinases which were not covered by the 36 probe pairs included in our assay (i.e., OXA-24, OXA-25, OXA-26, OXA-91, OXA-143). Further optimization of dMLA will likely improve assay sensitivity and reduce time of analysis and associated costs.

The dMLA workflow shares common features with MLPA assays[24], for instance the concept of annealing of two probes for subsequent ligation and multiplexed PCR. Recently, a combination of MLPA-like workflow with NGS has been used to screen samples for cancer-specific markers[14,15] and abnormal chromosomal markers in maternal blood[16]. Furthermore, conceptually similar approaches demonstrate target quantification as well as detection of single nucleotide polymorphisms[14–16,25]. Our approach also shares similarity with network analysis, examples of which have recently been used for degenerate primer design[13],

and highlights the use of in silico design and/or evaluation of PCR primers to improve target coverage and reduce laboratory work[26].

dMLA may be designed to quantify target gene regions in a sample and/or detect point mutations, for example gene mutations associated with increased antibiotic resistance. However, application of dMLA for reliable quantification will require optimizing the assay to control for potential differences in unique molecular counts attributable to factors beyond the number of cells per assay. Example factors include variable numbers of target genes per cell (chromosomal vs. plasmid gene targets) and variable efficiencies in probe-target binding affinity, ligation, and PCR amplification efficiency. The latter likely explain observed variation in unique molecular barcode counts amongst distinct probe pairs targeting the same gene within the same sample (Fig. 3, Supplementary Fig. 1). One potential approach to expand dMLA to quantification would be to incorporate standards within the assay, such as those applied in qPCR assays.

## Materials and methods

**dMLA probe design.** Probes for β-lactamase genes were designed to detect 18 β-lactamase resistance gene cluster sequences covering 1187 of 1557 (76%) of the β-lactamase genes in the Lahey dataset (accessed 26 July 2016)[3]. Two probe pairs were designed for each cluster to permit two-fold verification of cluster presence. To design probes, probe pair candidate sequences (40 bp) from all β-lactamase genes were generated using a sliding window function in R Statistical Software (v 3.5.1) (code available at https://github.com/manutamminen/dmla/). The universal set of 40-mers were used for a similarity search against all β-lactamase-encoding genes (implemented in *nsearch*;[17], and filtered using a set coverage algorithm (implemented in *epride*, at https://github.com/manutamminen/epride) to select the subset with maximum coverage of the Lahey sequence database (Supplementary Fig. 1). A subset of clusters were chosen for dMLA, based on clinical relevance and/or sequence coverage depth (Supplementary Fig. 1, Supplementary Table 1). The probe pair candidates within each cluster were further filtered to identify sets with shared annealing temperatures (<2 °C apart).

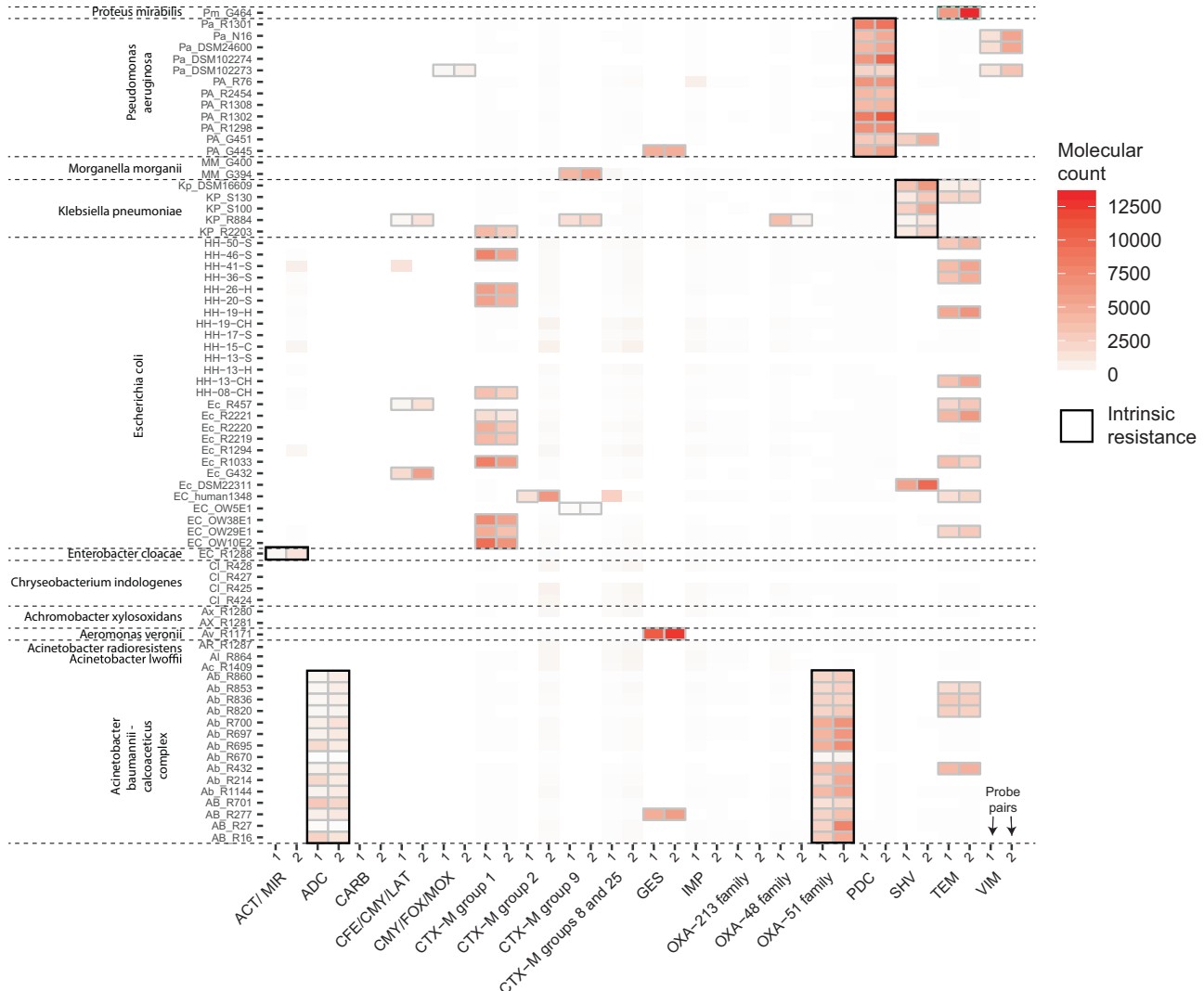

**Fig. 3 dMLA is used to screen 74 bacterial genomic DNA preparations (*y*-axis) using a mixture of 36 probe pairs (*x*-axis) capable of detecting β-lactamase encoding genes within 18 gene clusters.** Detection of a gene cluster is determined when the unique molecular target counts for two sets of unique probe pairs targeting the same cluster are significantly higher than background noise, denoted here by grey outlines (black in case of intrinsic resistance). Variation in unique molecular counts between the two sets of probe pairs for a given β-lactamase encoding gene family is driven by probe pair binding affinity and specificity and/or PCR efficiency.

**dMLA probe testing and validation**. Up to five candidate probe pairs per cluster were tested in singleplex PCR with synthetic template DNA of the probe pair target gene region (Microsynth AG, Balgach, Switzerland). Ligation proceeded in a Biometra T3000 thermocycler set to 94 °C for 10 min followed by 60 °C at 90 min using AmpLigase (5 units; Epicentre and Lucigen) and AmpLigase buffer, left and right ligation probes (40 nM each, Table S1), and template (100 fM, 1 and 10 pM for each probe pair) or a no template control (NTC) in a final volume of 25 µl. No-ligase controls (NLC) were tested without template and using 10 pM templates. Subsequent PCR (25 µl reaction) was performed by adding Phusion High-Fidelity DNA Polymerase (2 units; BioLabs), Phusion buffer (1x), dNTPs (final concentration 0.2 mM), forward primer (with sample-specific barcode denoted here as 'NNNNNNN'; 5′-TCTTTTCGCAGGCTGGAGCCCAGGTCTTCCTATNNNNN NNTGGGGCCCAATTTTCCGTGAC-3′, 400 nM), reverse primer (5′ - GAGTGCA CGCACACTCATTC-3′, 400 nM) and ligation product (1 µl) or NTC (RNAse/ DNAse free water) into a final volume of 25 µl. Thermocycling conditions were 98 °C for 30 s followed by 35 cycles of 98 °C for 5 s, 70 °C for 1 s, and 72 °C for 3 s with a final 72 °C for 5 min. PCR products (10 µl) were visualized in 1.5% agarose gels in TAE Buffer using 0.01% SYBR SAFE (Thermo Scientific) on a Biorad PowerPac 300 Gel Electrophoresis/Vilber E-Box VX-2 gel imager in comparison to 3 µl of 100 bp standard ladder (Promega). Probe pairs were deemed unsuccessful if no band was visible at 140 bp or if a band was visible at 140 bp in the NTC or NLC. Approximately half of all probe pairs tested were unsuccessful, with detection of a band in the NLC as the most common cause.

The dMLA was tested with 74 bacterial isolates for 18 clusters using two probe pairs for each cluster (Fig. 2). The bacterial isolates included 60 isolates hosting known β-lactamase genes previously characterized by Patrice Nordmann's lab (University of Fribourg, Fribourg, Switzerland, personal communication) and 14 *Escherichia coli* isolates from Bangladesh[27,28]. The bacterial genomic DNA was extracted by boiling at 100 °C for 1 h. Ligation and PCR proceeded as described, with a probe pair mix totalling 1 µM, and 2 probe pairs per cluster for 18 clusters. The PCR products were quantified using a Qubit dsDNA high sensitivity kit (Invitrogen, CA, USA) and pooled based on concentration prior to Illumina MiSeq sequencing (Eurofins) with 150 bp paired end reads, which was chosen to ensure complete doubly confirmed coverage of the sequenced area (138 bp).

**Sequence data analysis**. Paired-end merging, quality filtering, and read mapping of the FASTQ output data from Illumina HiSeq was performed using *nsearch*[17]. Reads were then mapped to reference databases to identify associated β-lactamase-encoding gene clusters and sample IDs (bacterial isolate). For each combination of cluster and sample IDs, the unique molecular barcodes (Fig. 1a) were counted. The detection limit was calculated based on experimental background noise estimated from NTCs and/or susceptible bacterial isolates with no resistant genes. Sampling-corrected signals exceeding the sum of the mean with three times the standard deviation of the blank experiments in both replicates were considered reliable. The sequence data (PRJNA531165) is available at SRA. For further detail, the workflow is available online at https://github.com/manutamminen/dmla with the reference data set and annotation available at: https://github.com/manutamminen/dmla/ blob/master/Allele-dna.fa and https://github.com/manutamminen/dmla/blob/ master/Allele.tab, respectively. β-lactamase-encoding genes identified using dMLA

that are not intrinsic resistance genes were confirmed using previously published multiplexed PCR primers[8].

**Statistics and reproducibility.** The dMLA detection limits were defined based on false positive signals in triplicate, independently prepared blank samples. From these triplicate samples, the mean and standard deviation of the number of unique molecules arising as false positive signals were calculated for each probe target independently. The detection limit was defined as the sum of the mean and three times the standard deviation, equivalent to 99.7% of the distribution assuming it is normally distributed. Any signal from the samples above this limit was considered statistically significant, which was used to differentiate between detection and non-detection.The analyses are available at https://github.com/manutamminen/dmla.

The data were replicated using two independent sequencing runs, one with 35 PCR cycles (presented here) and one with 40 PCR cycles. The detect/non-detect classification was identical.

**Reporting summary.** Further information on research design is available in the Nature Research Reporting Summary linked to this article.

## Data availability

The datasets generated during and/or analysed during the current study are available in the following repositories. DNA sequencing data is available at Sequence Read Archive (SRA; https://www.ncbi.nlm.nih.gov/sra) under accession PRJNA531165. β-lactamase sequences, their metadata and the probe designs are available at https://github.com/manutamminen/dmla.

## Code availability

All source code for preparing the dMLA probes, processing the DNA sequencing data, visualizing the results and performing the statistical tests is publicly available at https://github.com/manutamminen/dmla.

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

## Acknowledgements

This work was funded by the Bill and Melinda Gates Foundation through grant OPP1151041 and the Swiss National Science Foundation (SNSF) through grant OP157065. The funding agencies had no role in the study design, data collection, interpretation of the results, or submission of the work for publication. Thanks to Dr. Laurent Poirel, Dr. Patrice Nordmann, and laboratory for providing bacterial isolates, and Dr. Ana Karina Pitol for assistance in the laboratory.

## Author contributions

M.T. and T.R.J. conceived of the project, planned experiments, supervised the project, and wrote the paper. J.S., L.C., H.S., and R.L. planned and conducted experiments. M.C.M. planned experiments. M.T. and S.S. developed the bioinformatic tools and analyzed the data. All authors edited the paper.

## Competing interests

Authors M.T., T.R.J., J.S., L.C. and H.S. hold a patent (Patent # CA3072650A1) on a similar method for detection and quantification of genetic targets in human, animal, and environmental samples. The workflow presented in this study is distinct from that covered by the patent, though it is conceptually related. R.L., S.S and M.C.M. declare no competing interests.
