## [Peer Review File · Communications Biology]

Reviewers' comments:

Reviewer #1 (Remarks to the Author):

The manuscript 'Digital Multiplex Ligation Assay for Highly Multiplexed Screening of β -Lactamase-encoding Genes in Bacterial Isolates' describes the development and evaluation of a modern and robust method for identification of beta-lactamase genes in various clinical isolates. The manuscript is clearly written, and the developed method could be implemented in laboratories with next-generation sequencing facilities. There are just a few comments and suggestions that are desirable to be addressed by the Authors:

Major:

1. Figure 3 represents the main findings of the study, but there is a mess with two colour palettes used. The reader could not compare directly molecular counts of 'intrinsic' resistance genes and 'confirmed' genes. Consider the use of the same gradient with different outlines.
2. The Authors should discuss the interconnection between molecular count obtained by the described method and copy-number of genes per chromosome in clinical samples. If the sample does not contain an intrinsic gene, which molecular count is proportional to genomic equivalents, then the plasmid-borne genes cannot be accurately counted (*M. morgani* and *A. veronii* cases). There is also a difference between the two probes for one gene in some samples (Figure 3). For example, the ratio of TEM probes in *P. mirabilis* and *E. coli* seems to be different. What is the cause of this observation?

Minor comments:

1. It is not entirely clear how the NGS data was processed. Particularly, what sequences were used as references for mapping?
2. How many samples could be analyzed simultaneously in the single NGS run?
3. References (Gorecki et al. 2019) (Saare et al. 2019) (Schmid et al. 2018) in the main text are absent in the References section.
4. Software linked to <https://github.com/manutamminen/dmla/> is not available (as of Nov, 11).

Reviewer #2 (Remarks to the Author):

This manuscript describes a proof-of-principle study to demonstrate the feasibility of detecting beta-lactamase-coding genes in bacterial isolates using a probe hybridization & ligation based assay with next-generation sequencing readout. The text is easy-to-follow, the figures are of good quality and the related literature is properly cited.

Specific comments:

- What is the sensitivity and quantitative range of the assay? Have the authors analyzed a dilution series?
- Have the authors checked the performance of the assay to detect targets in mixtures of bacterial isolates?
- Why were different molecular counts detected for the matching half probes if those were amplified together after ligation? (coloring in Fig 3) How were these sometimes quite substantial differences handled during the analysis?
- It would be useful to include the information why exactly 180bp paired-end sequencing was applied?
- Recommendations should be included what sequencing power (read number) should be applied/achieved to ensure an efficient detection of ARGs. I presume the number of targets has a profound influence in this regard.
- Some info should be included how exactly the isolates were previously characterized, i.e. what kind of methods validated the findings of dMLA?

- Fig 1A: 'Molecule barcode' vs 'Molecular barcode'

- Fig 1B and 1C: It would be better to indicate the length of the barcodes instead of including specific sequences. The info in its current form is a bit misleading.

Reviewer #3 (Remarks to the Author):

The manuscript reports the development of a multiplexed gene analysis method called digital multiplex ligation assay (dMLA). This method was applied for the detection of antibiotic resistance genes in 74 bacterial genomes in DNA preparations using 36 probe pairs. These probes were designed in-silico with degenerate regions to cover 1187 beta-lactamase genes.

Overall, the method seems very versatile and the massive multiplexing capabilities are certainly of interest for many applications beyond the screening of antibiotic resistance. Concerning this particular application, the results are interesting and certainly show the potential of the technique in the context of screening antibiotic resistance, which urgently demands cost effective, accurate and rapid analytics.

On the other hand, I have a number of major concerns with the (1) novelty/impact, (2) benchmarking, (3) characterisation of the technique and; (4) interpretation/visualization of the acquired data. These are listed below in detail:

1- An overview of the state of the art is given in the introduction highlighting the lack of scalability and high cost as major limiting factors, which can be overcome by the reported method. While this is clear for all methods in lines 24-26, it is not clear what are the key differences between dMLA reported here and the previously reported digital MLPA. The authors simply vaguely mention that the technique is closely related but distinct and provide a number of features in lines 37 to 39 which are also shared with digital MLPA, also having scalability up to the thousands of genes (making lines 7-8 on the Abstract not entirely correct). Overall, the novelty of the reported technique has to be better highlighted.

2- The benchmarking of this technique relative to the state of the art should be improved by comparing key figures of merit (e.g. in the form of a table) such as the scalability, minimum requirements of gDNA mass/concentration, cost and total time of analysis.

3- Following the previous point, the characterisation of dMLA should be improved to include minimum requirements of starting genomic material (limit of detection) as performed for example by the authors of the reference in lines 205-208 reporting digital MLPA, as well as a validation of linearity vs concentration of target gene and detection cut-off.

4- Concerning the results shown in Figure 3, the authors state that no false positives or negatives were found. This is not clear from the way the results are presented. There seems to be a false negative for EC_OW5E1 since the count number seems to be lower than other genes which were not considered positive (neither intrinsic nor confirmed). Furthermore, some of the intrinsic genes seem to be almost "white" with no counts (false negative). Also, some genes with counts having a significant color intensity are found neither intrinsic nor confirmed (potential false positives). In any case, as mentioned on the previous point, the detection limit must be clearly highlighted. Furthermore, quantitative information of Figure 3, besides the color coding alone, should also be provided as supplementary information.

Response to the reviewer comments of the manuscript COMMSBIO-19-1511 entitled “Digital Multiplex Ligation Assay for Highly Multiplexed Screening of β -Lactamase-encoding Genes in Bacterial Isolates”.

We thank the editor and the reviewers for the comments and suggestions which we believe have strengthened our manuscript. The following lists the point- by-point responses to the comments. Editor and reviewer comments are numbered, responses are indented below the comments, excerpts from the manuscript are in quotation marks, with new updates emboldened.

Editor’s Comments:

E.1. However, please bear in mind that we will be reluctant to approach the referees again in the absence of major revisions especially concerned with i) addressing if the method can be used to detect targets using input of bacterial isolate mixture (R2),

The assay can likely be used for more complex mixtures beyond the current implementation, but we have not yet tested this. The current assay implementation goal is to identify, for large sets of single bacterial isolates, multiple target genes. As such, the current workflow describes ligation and amplification as performed on single bacterial isolates, which are then mixed into one batch of multiple (74) bacterial isolates prior to sequencing. The samples are barcoded during amplification, which is a necessary step to link the target genes back to the original isolates.

That said, we believe the sample complexity of a single bacterial isolate would be similar to the sample complexity of pooled bacterial isolates, suggesting dMLA would work in this scenario. See response to Reviewer Comment 2.2.

E.2. addressing the discrepancy in the molecular counts (R2),

The apparent discrepancy arises due to limitations of using a heat map for visualization of the unique molecular bar counts that vary highly between various probe pairs. We have remedied this by providing the data in a different format and improving visualization of the existing heat map. Please see response to Reviewer Comments R1.1, R.1.2, and most notably R2.3

E.3. clarifying the novelty of the work (R3),

The method we describe integrates ligation-based assays with NGS to multiplex and parallelize detection of a large cluster of antibiotic resistance genes. The workflow is conceptually similar to a couple novel studies that were recently published (Benard-Slagter et al; Kosztolanyi et al; Teder et al), and is optimized for detection of antibiotic resistance genes in bacterial isolates. Antibiotic resistance gene detection requires designing probes which cover a large diversity of target genes, and providing a design workflow for such probes is another point of novelty of the manuscript.

E.4. further bench-marking the method as suggested by R3,

We agree that our method should be benchmarked against other available methodologies at some point in future. However, we are publishing here the first proof-of-the-concept of the method for

antibiotic resistance gene detection and therefore would not find it entirely fair to compare its performance to commercially available methods which have gone through extensive optimization and streamlining. Therefore we find it sufficient here to present the results as they are and also add, for review purposes a note about the scalability, presented as a response to Comment 1.4.

E.5. and addressing the point raised about true false negatives/positives (R3) among other clarification points raised by R1, R2, and R3.

We provide a new Figure S01 which represents the number of unique molecular counts as bar charts and their relation to the algorithmic detection thresholds. This figure also points out those responses which are not consistently observed across both probe pairs, and unambiguously shows the lack of false positives or negatives.

Reviewers' comments:

Reviewer 1:

R1.1 The manuscript 'Digital Multiplex Ligation Assay for Highly Multiplexed Screening of β -Lactamase-encoding Genes in Bacterial Isolates' describes the development and evaluation of a modern and robust method for identification of beta-lactamase genes in various clinical isolates. The manuscript is clearly written, and the developed method could be implemented in laboratories with next-generation sequencing facilities. There are just a few comments and suggestions that are desirable to be addressed by the Authors

Major:

Figure 3 represents the main findings of the study, but there is a mess with two colour palettes used. The reader could not compare directly molecular counts of 'intrinsic' resistance genes and 'confirmed' genes. Consider the use of the same gradient with different outlines.

Thanks for this comment, we have updated Figure 3 to improve visualization. Most notably, we have replaced outline colors with a light gray color to differentiate detect/non-detects, and used a black outline for groups of intrinsic genes, which provides a more clear relative visualization of the number of unique molecular barcodes, but still allows the indicator of intrinsic vs. PCR-confirmed genes to be clearly distinct. We have also included a different visualization using bar charts (instead of a heat map) in the supplemental (Figure S01), to allow better comparison of unique molecular bar counts across probe pairs. We now reference Figure S01 whenever we reference Figure 3 as the figures rely on the same data.

R1.2. The Authors should discuss the interconnection between molecular count obtained by the described method and copy-number of genes per chromosome in clinical samples. If the sample does not contain an intrinsic gene, which molecular count is proportional to genomic equivalents, then the plasmid-borne genes cannot be accurately counted (*M. morganii* and *A. veronii* cases). There is also a difference between the two probes for one gene in some samples (Figure 3). For example, the ratio of TEM probes in *P. mirabilis* and *E. coli* seems to be different. What is the cause of this observation?

The reviewer highlights additional considerations needed before the current implementation of dMLA (described here for presence/absence detection of genes within bacterial isolates) can be expanded to target gene quantification. We expect the method is also adaptable to quantification in the future, and we reference other similar methods that allow target gene quantification:

“Furthermore, conceptually similar approaches demonstrate target quantification as well as detection of single nucleotide polymorphisms (Kosztolányi et al. 2018; Benard-Slagter et al. 2017; Teder et al. 2018; Saare et al. 2019). dMLA may therefore be designed to quantify target gene regions in a sample and/or detect point mutations, for example gene mutations associated with increased antibiotic resistance.”

However, as the reviewer states, there are multiple factors influencing the number of unique molecular counts used to estimate gene presence/absence. These factors need to be accounted for and optimized before dMLA is able to be implemented reliably for target gene quantification. Examples include what the reviewer mentioned (copies of target per cell) as well as other factors (primer binding, ligation, and pcr amplification efficiency as examples). The latter factors likely account for the reviewer’s observation of a different ratio of TEM probes observed in *P. mirabilis* as compared to *E. coli*. We have included a new section on these factors in the discussion following the above to highlight and address the reviewers concerns:

“However, application of dMLA for reliable quantification will require optimizing the assay to control for potential differences in unique molecular counts attributable to factors beyond the number of cells per assay. Example factors include variable numbers of target genes per cell (chromosomal vs. plasmid gene targets) and variable efficiencies in probe-target binding affinity, ligation, and pcr amplification efficiency. The latter likely explain observed variation in unique molecular barcode counts amongst distinct probe pairs targeting the same gene within the same sample (Figure 3). One potential approach to expand dMLA to quantification would be to incorporate standards within the assay, such as those applied in QPCR assays.”

Please also refer to Reviewer Comment 2.3 for additional relevant comments about this topic.

R1.3. Minor comments: It is not entirely clear how the NGS data was processed. Particularly, what sequences were used as references for mapping?

We thank the reviewer for this important point, we should have made the workflow and repositories public earlier. In response, we have updated our external resources to provide substantially more detail on the processing steps, provided at <https://github.com/manutamminen/dmla/blob/master/workflow.org>. The reference set is <https://github.com/manutamminen/dmla/blob/master/Allele-dna.fa> and annotation <https://github.com/manutamminen/dmla/blob/master/Allele.tab>. The manuscript has been updated in the Materials and Methods to include:

“The sequence data (PRJNA531165) is available at SRA. For further detail, the workflow is available online at:

<https://github.com/manutamminen/dmla/blob/master/workflow.org>, with the reference data set and annotation available at: <https://github.com/manutamminen/dmla/blob/master/Allele-dna.fa> and <https://github.com/manutamminen/dmla/blob/master/Allele.tab>, respectively.”

R1.4 How many samples could be analyzed simultaneously in the single NGS run?

This is a question of sequencing depth which depends on the sequencing platform used. For instance, Illumina MiSeq Reagent Kit v2 Micro provides a depth of 30 million paired-end reads. With the average sequencing depth within our samples, this translates into up to 249 samples on a single run. On the other hand, with the depth of 20 billion paired-end reads on NovaSeq S4, this translates into an average of ~166000 samples on a single run.

R1.5 References (Gorecki et al. 2019) (Saare et al. 2019) (Schmid et al. 2018) in the main text are absent in the References section.

This has now been fixed, and the appropriate reference has also been included in the main text.

R1.6 Software linked to <https://github.com/manutamminen/dmla/> is not available (as of Nov, 11).

This has now been fixed.

Reviewer #2

R2.1. This manuscript describes a proof-of-principle study to demonstrate the feasibility of detecting beta-lactamase-coding genes in bacterial isolates using a probe hybridization & ligation based assay with next-generation sequencing readout. The text is easy-to-follow, the figures are of good quality and the related literature is properly cited. Specific comments: What is the sensitivity and quantitative range of the assay? Have the authors analyzed a dilution series?

Method is used for presence/absence detection in bacterial isolates where sensitivity/qr are not the focus. However, we have performed dilution series experiments from which molar detections limits can be estimated, and present those results here for review purposes:

Table R1. Detection limits and R2 and P-values for the probe pairs and targets.

Family	Probe pair	R2	pval	Detection limit (attomoles / μ l)
TEM	2	0.999	< 0.001	20.59
OXA-51 family	1	0.758	0.024	50.353
OXA-51 family	2	0.973	< 0.001	47.38

VIM	1	0.996	< 0.001	47.818
VIM	2	1	< 0.001	10.056
GES	1	0.991	< 0.001	27.143
GES	2	0.987	< 0.001	16.282
CMY/FOX/MOX	1	0.996	< 0.001	13.485
CMY/FOX/MOX	2	0.985	< 0.001	17.167
CARB	1	0.965	< 0.001	12.586
CARB	2	0.965	< 0.001	4.271
SHV	1	0.102	0.538	207.687
SHV	2	0.956	0.001	26.708
CMY	2	0.978	< 0.001	14.63
PDC	2	0.991	< 0.001	0.149
CTX-M group 1	1	0.991	< 0.001	5.674
CTX-M group 1	2	0.841	0.01	10.184
ADC	1	0.851	0.009	24.401
ACT/ MIR	1	0.994	< 0.001	24.755
ACT/ MIR	2	0.967	< 0.001	26.314

CTX-M group 9	2	0.958	0.001	202.87
---------------	---	-------	-------	--------

R2.2. Have the authors checked the performance of the assay to detect targets in mixtures of bacterial isolates?

No, the assay has currently been restricted to detection of targets in single bacterial isolates pooled after ligation and amplification but before sequencing. However, we were able to detect multiple resistance genes in genomic preps of bacterial isolates, thus confirming the capability of the method to detect multiple targets against a complex background. We therefore do not expect pooling of multiple bacterial isolates to affect the performance of the assay. This is due to the fact that the genomic DNA from single bacterial isolates already provides a very complex background for the detection. We have updated the description of this in the results/discussion section to also include pooled bacterial isolates:

“Notably, the approach is readily adaptable to the detection of alternative targets and/or alternative sample types such as screening samples (pooled bacterial isolates, blood, saliva, soil, stool, water) for markers of cancer (Benard-Slagter et al. 2017; Kosztolányi et al. 2018),”

R2.3. Why were different molecular counts detected for the matching half probes if those were amplified together after ligation? (coloring in Fig 3) How were these sometimes quite substantial differences handled during the analysis?

Please see response to Reviewer Comments 1.1. and 1.2. Further, the molecular counts represent two pairs of half probes. For each target gene, two distinct 40bp regions were detected using two pairs of half probes (four half probes total). Although theoretically the number of initial target gene regions should be equal amongst the two sets of half probes, differences in probe pair binding affinity and specificity, and/or PCR amplification efficiency likely account for differences in molecular counts. For specificity, this can be clearly observed by the variation in the number of unique molecular counts within the control samples, with some probe pairs (CTX-M-group 2, 2nd probe pair) having more background noise (non-specific binding within controls) than their matched probe pair.

The algorithm for detecting genes based on the number of unique molecular barcodes requires that the number of unique molecular barcodes detected is greater than the the average number of unique molecular barcodes plus three standard deviations in the corresponding negative control for both primer pairs. Combinations of probe pairs that do not meet this threshold (in either or both probe pairs individually) are deemed undetected.

We have updated the manuscript to include this information in the Figure 3 caption:

Variation in unique molecular counts between two sets of probe pairs for a given beta-lactamase encoding gene family is driven by probe pair binding affinity and specificity and/or PCR efficiency.

R2.4 It would be useful to include the information why exactly 180bp paired-end sequencing was applied?

We designed each half-probe to include a 20 bp region of the target gene, or half of the total 40 bp targeted gene region. This is sufficiently large to ensure specificity. The half-probes also contain molecular barcodes (molecule or sample-specific) and overhangs for subsequent PCR amplification. Taken together, the sequenced area of the ligated, amplified product is 138 bp. Thus, this probe length is a good match for a certain sequencing length available on MiSeq. The Reviewer's comment also highlighted an error in the manuscript: we used 150 bp paired end reads (not 180 bp). We have updated the manuscript by adding the following to the Methods:

The PCR products were quantified using a Qubit and pooled based on concentration prior to Illumina MiSeq sequencing (Eurofins) with **150 bp paired end reads, which was chosen to ensure complete, doubly confirmed coverage of the sequenced area (138 bp).**

R2.5 Recommendations should be included what sequencing power (read number) should be applied/achieved to ensure an efficient detection of ARGs. I presume the number of targets has a profound influence in this regard.

This is a question of sequencing depth which depends on the sequencing platform used and is discussed in response to Reviewer comment 1.2.

R2.6. Some info should be included how exactly the isolates were previously characterized, i.e. what kind of methods validated the findings of dMLA?

The isolates were chosen based on prior knowledge provided by the source lab (Nordman lab) who previously identified genes using molecular methods (PCR and/or sequencing). Our lab further confirmed all detects by PCR or identified them as intrinsic genes based on references from the literature. A subset of these isolates with intrinsic genes were also confirmed by PCR in our lab.

We state this in the manuscript:

The bacterial isolates included 60 isolates hosting **known β -lactamase genes previously characterized by Patrice Nordmann's lab (University of Fribourg, Fribourg, Switzerland, personal communication)** and 14 *Escherichia coli* isolates from Bangladesh (Montealegre et al. 2018).

and

β -lactamase-encoding genes identified using dMLA that are not intrinsic resistance genes were confirmed using previously published multiplexed PCR primers (Dallenne et al. 2010).

R2.7 Fig 1A: 'Molecule barcode' vs 'Molecular barcode'

We have updated this.

R2.8 Fig 1B and 1C: It would be better to indicate the length of the barcodes instead of including specific sequences. The info in its current form is a bit misleading.

We have updated this.

Reviewer #3

R3.1. The manuscript reports the development of a multiplexed gene analysis method called digital multiplex ligation assay (dMLA). This method was applied for the detection of antibiotic resistance genes in 74 bacterial genomes in DNA preparations using 36 probe pairs. These probes were designed in-silico with degenerate regions to cover 1187 beta-lactamase genes.

Overall, the method seems very versatile and the massive multiplexing capabilities are certainly of interest for many applications beyond the screening of antibiotic resistance. Concerning this particular application, the results are interesting and certainly show the potential of the technique in the context of screening antibiotic resistance, which urgently demands cost effective, accurate and rapid analytics.

On the other hand, I have a number of major concerns with the (1) novelty/impact, (2) benchmarking, (3) characterisation of the technique and; (4) interpretation/visualization of the acquired data. These are listed below in detail

We appreciate these comments.

R3.2- An overview of the state of the art is given in the introduction highlighting the lack of scalability and high cost as major limiting factors, which can be overcome by the reported method. While this is clear for all methods in lines 24-26, it is not clear what are the key differences between dMLA reported here and the previously reported digital MLPA. The authors simply vaguely mention that the technique is closely related but distinct and provide a number of features in lines 37 to 39 which are also shared with digital MLPA, also having scalability up to the thousands of genes (making lines 7-8 on the Abstract not entirely correct). Overall, the novelty of the reported technique has to be better highlighted.

The dMLA method employed here is very similar to the other listed techniques in terms of the laboratory workflow. However, such laboratory workflow in itself is very novel since, to our knowledge, only a few articles exist so far where it has been employed (Benard-Slagter et al; Kosztolanyi et al; Teder et al). Our study is the first one where this workflow is applied to the detection of antibiotic resistance genes.

We reference these studies the manuscript:

Recently, a closely related approach, digitalMLPA, was applied for the detection of copy number alterations in T- and B-cell lymphoblastic leukemia (Kosztolányi et al. 2018; Benard-Slagter et al. 2017) and multiple myeloma (Kosztolányi et al. 2018; Benard-Slagter et al. 2017). **A closely related approach, TAC-seq, was also recently introduced to improve the sensitivity of non-invasive pre-natal trisomy testing (Teder et al. 2018).**

Recently, a combination of MLPA-like workflow with NGS has been used to screen samples for cancer-specific markers (Benard-Slagter et al. 2017; Kosztołányi et al. 2018) **and abnormal chromosomal markers in maternal blood (Teder et al 2018).**

An important part of the novelty of the current study is the primer design workflow which utilises BLAST-like similarity searches and maximum coverage algorithm to detect the optimal probe binding sites within the resistance gene data set. Using this design workflow we were able to design 36 degenerate oligonucleotide probes which cover 1187 genes of the total pool of 1557 resistance genes. Thus, we consider our scalability claim valid.

R3.3- The benchmarking of this technique relative to the state of the art should be improved by comparing key figures of merit (e.g. in the form of a table) such as the scalability, minimum requirements of gDNA mass/concentration, cost and total time of analysis.

We appreciate this comment, and we agree that our approach eventually needs to be benchmarked against the state-of-the-art to fully demonstrate merit. However, at this point the method has been completed and demonstrated but not optimized, certainly not for commercial applications. Therefore, estimates on many of the benchmarks discussed by the review would skew to overestimates relative to the potential of the method. For example, the total time of analysis could be dramatically improved with in-house sequencing. Comparison to other (commercialized) methods would not reflect the full potential of this method. Once the method is further implemented or streamlined, we will be able to more accurately bench mark the methods. See response to Reviewer Comment 1.4 for questions of scalability. We have included the following to state this clearly:

Further optimization of dMLA will likely improve assay sensitivity and reduce time of analysis and associated costs.

R3.4- Following the previous point, the characterisation of dMLA should be improved to include minimum requirements of starting genomic material (limit of detection) as performed for example by the authors of the reference in lines 205-208 reporting digital MLPA, as well as a validation of linearity vs concentration of target gene and detection cut-off.

Please refer to Reviewer Comment 2.1.

R3.5- Concerning the results shown in Figure 3, the authors state that no false positives or negatives were found. This is not clear from the way the results are presented. There seems to be a false negative for EC_OW5E1 since the count number seems to be lower than other genes which were not considered positive (neither intrinsic nor confirmed). Furthermore, some of the intrinsic genes seem to be almost "white" with no counts (false negative). Also, some genes with counts having a significant color intensity are found neither intrinsic nor confirmed (potential false positives). In any case, as mentioned on the previous point, the detection limit must be clearly highlighted. Furthermore, quantitative information of Figure 3, besides the color coding alone, should also be provided as supplementary information.

Thanks for this important point. Detect/non-detect is determined algorithmically, as described in response to Reviewer Comment 2.3. For some samples (EC_OW5E1), the number of unique molecular counts is significantly higher than the background noise that is considered a positive signal. However, because

there is substantial variation amongst the unique molecular counts across all sample/probe pairs, it is difficult to visually see detects in the heat map provided in Figure 3 (as the reviewer states). We have remedied this by providing a new Figure S01 in the supplementary material, which represents the number of unique molecular counts as a bar chart, with a vertical line representing the algorithmic detection threshold. We reference this new figure alongside Figure 3, which we have also updated in response to Reviewer Comment 1.1 such that the color coding scheme makes the heat map visualization more clear.

REVIEWERS' COMMENTS:

Reviewer #1 (Remarks to the Author):

The authors addressed the main concerns; the revised version of the manuscript appears to be good. This proof-of-concept study represents a sufficient advance beyond the previously described methods for detection of β -lactamase-encoding genes in bacterial isolates. Moreover, even the developed epride algorithm for primer design is worth publishing.

Previous studies utilizing a similar approach were correctly cited.

Another interesting point raised by the other reviewers was about bacterial mixtures. It is an interesting scientific question, but I suppose it is not clinically important to obtain the result about the particular pathogen and its corresponding resistant profile. The joined data on resistance of two or more pathogenic bacteria would be used for treatment.

Minor corrections

1. Workflow at <https://github.com/manutamminen/dmla/blob/master/workflow.org> is not available; however, it is not necessary, since the reference fasta file and mapping script were published.

2. Page 7: 'available as SRA.'

Reviewer #3 (Remarks to the Author):

The authors adequately addressed all concerns raised by the reviewers. The manuscript is now acceptable for publication.